# A Review on Additive Manufacturing Methods for NiTi Shape Memory Alloy Production

**DOI:** 10.3390/ma17061248

**Published:** 2024-03-08

**Authors:** Kristýna Kubášová, Veronika Drátovská, Monika Losertová, Pavel Salvetr, Michal Kopelent, Filip Kořínek, Vojtěch Havlas, Ján Džugan, Matej Daniel

**Affiliations:** 1Department of Mechanics, Biomechanics and Mechatronics, Faculty of Mechanical Engineering, Czech Technical University in Prague, Technická 4, 160 00 Prague, Czech Republicveronika.dratovska@fs.cvut.cz (V.D.); 2Department of Materials Engineering and Recycling, Faculty of Materials Science and Technology, VSB—Technical University of Ostrava, 17. listopadu 2172/15, 708 00 Ostrava, Czech Republic; mlosertova@vsb.cz (M.L.); michal.kopelent@vsb.cz (M.K.); 3COMTES FHT a.s., Průmyslová 995, 334 41 Dobřany, Czech Republicjan.dzugan@comtesfht.cz (J.D.); 4Department of Orthopaedics and Traumatology, 2nd Faculty of Medicine, Charles University and Motol University Hospital, 150 06 Prague, Czech Republic; filip.korinek@fnmotol.cz (F.K.); vojtech.havlas@lfmotol.cuni.cz (V.H.)

**Keywords:** NiTi alloy, additive manufacturing, shape memory alloy

## Abstract

The NiTi alloy, known as Nitinol, represents one of the most investigated smart alloys, exhibiting a shape memory effect and superelasticity. These, among many other remarkable attributes, enable its utilization in various applications, encompassing the automotive industry, aviation, space exploration, and, notably, medicine. Conventionally, Nitinol is predominantly produced in the form of wire or thin sheets that allow producing many required components. However, the manufacturing of complex shapes poses challenges due to the tenacity of the NiTi alloy, and different processing routes at elevated temperatures have to be applied. Overcoming this obstacle may be facilitated by additive manufacturing methods. This article provides an overview of the employment of additive manufacturing methods, allowing the preparation of the required shapes of Nitinol products while retaining their exceptional properties and potential applications.

## 1. Introduction

Shape memory alloys (SMAs) are a class of “smart materials” with the remarkable ability to recover their original shape after significant deformation triggered by specific temperature changes. While several alloys exhibit this phenomenon, including Cu-Zn-Al, Co-Ni-Al, Cu-Al-Ni, Ni-Mn-Ga, and Ti-Cu [1,2,3,4], nickel–titanium (NiTi), also known as Nitinol, has garnered the most attention due to its unique properties. Discovered in 1965 by Buehler and Wang [5], Nitinol is typically an equiatomic intermetallic compound containing 50 atomic percent (55 weight percent) Ni, and the remainder is Ti. However, modern Nitinol alloys encompass a broader range of binary compositions based on these elements. 

Advances in science and technology have driven the widespread adoption of Nitinol-based alloys in various industries. These alloys find applications in specialized engineering domains, including temperature sensors, thermoelectric actuators, and thermomechanical connectors, particularly in the automotive, aerospace, and consumer goods sectors [6,7]. 

Beyond its shape memory effect (SME) and superelasticity (SE), Nitinol exhibits excellent biocompatibility despite concerns regarding potential nickel ion release [5,8]. Its corrosion resistance rivals 300 series stainless steel and titanium alloys [9] and can be further enhanced through surface treatments that form protective layers [10,11]. Additionally, Nitinol demonstrates good wear resistance within the human body environment [12]. These combined characteristics make it highly suitable for medical devices, propelling its extensive use in biomedicine [8,13]. Nitinol’s first medical application emerged in the 1970s with superelastic Nitinol braces [14]. Since then, its reach has expanded to encompass various medical devices, including stents, extractors, artificial heart valves, constrictor fixators, nails, and plates. However, current applications often involve simpler shapes and smaller product sizes. This limitation is expected to be overcome by advancements in 3D modeling, shape optimization, and additive manufacturing (AM) technologies. By combining these methods, we can create products from diverse materials, including metal powders or strings, with intricate structures and varying sizes [7]. 

Unlike conventional alloys, the AM of smart materials like Nitinol presents additional complexities. This concept involves post-printing treatments that utilize external stimuli like heat or stress to shape or deform the 3D-printed parts. The time change elevates the technology to a new level, aptly named 4D printing [6,7,14]. While AM offers undeniable advantages, it also presents several drawbacks that can significantly alter the final properties of SMAs. The quality of the Nitinol product, its phase composition and microstructure, and its mechanical and physicochemical properties all depend heavily on optimized processing parameters [7]. This article delves into the current experience with AM technologies used for processing complex Nitinol structures, discussing the influence of these parameters on the final outputs and potential applications.

## 2. Production of NiTi and Methods of Additive Manufacturing

Nickel–titanium (NiTi) alloys are typically manufactured using established methods like vacuum melting and casting followed by forming, machining, or powder metallurgy [15,16,17,18]. While melting and casting remain prevalent, these techniques can introduce significant impurities, particularly oxygen and carbon, into the final product. Additionally, the melting process can induce micro- and macro-defects within the structure. Furthermore, it can lead to the formation of undesirable secondary phases with low melting temperatures, compromising mechanical properties [10,19]. Achieving precise shapes with good surface quality often requires additional machining, which poses challenges for NiTi due to its high ductility and resistance to deformation, resulting in significant tool wear. Laser cutting, electrical discharge machining, photochemical etching, and water jet cutting are alternative methods but have limitations in producing complex shapes [10,15,19,20]. Despite these drawbacks, conventional forming and machining remain dominant approaches for NiTi alloy production, at least for now [7,15,19,21].

Recent years have witnessed a surge in the application of additive manufacturing (AM) for fabricating NiTi components with diverse bulk and porous geometries [22]. This technology offers significant advantages, especially in medical applications. AM enables the precise control of mechanical properties, tailoring them to match the replaced tissue for improved biocompatibility. Additionally, it allows for the production of customized medical implants and the optimization of topology to reduce weight and elastic modulus, providing a novel approach to clinical treatment beyond just orthopedics [19,20,22,23]. Metal AM processes can be categorized based on several factors, as depicted in Figure 1. Key criteria include material distribution (powder bed or nozzle, Figure 2) and the energy source for fusion (laser, electron beam, or arc). We provide an overview of specific AM methods employed for NiTi alloy fabrication in the following sections.

### 2.1. Powder Bed Fusion—Laser Beam 

The most common [7,25,26,27,28,29,30,31,32,33] method to produce 3D-printed products from NiTi is the laser powder bed fusion (L-PBF) method. Belonging to the powder bed fusion (PBF) family with laser beam irradiation, also denoted as PBF-LB, L-PBF relies on a fundamental principle: selectively melting the desired shape layer-by-layer within a powder bed using a focused laser beam (Figure 3). After each layer’s solidification, a fresh layer of powder is deposited and subsequently melted. This intricate process typically occurs within a preheated chamber under controlled temperature and atmosphere, often involving an inert gas depending on the specific material and processing parameters. Notably, variations in these parameters significantly influence the final product’s properties, encompassing its microstructure, mechanics, and even physicochemical characteristics [32,33,34,35,36,37,38,39,40,41,42,43].

### 2.2. Powder Bed Fusion—Electron Beam 

Electron beam powder bed fusion (PBF-EB), also known as selective electron beam melting (SEBM), emerged from the Swedish company Arcam AB under the trade name electron beam melting (EBM). Similarities exist with PBF-LB (or Selective Laser Melting—SLM), as both utilize a system comprising powder hoppers, a rake, a printing platform, and an energy source to melt the powder (Figure 4). However, the crucial difference lies in the energy source itself: PBF-LB employs a laser beam, while PBF-EB harnesses the power of an electron beam. This necessitates a high vacuum environment within the system to maintain beam stability and power [44].

EBM works by accelerating electrons emitted from a tungsten filament to high speeds using high voltage. These electrons are then focused by lenses into a concentrated, high-energy beam. For materials like Nitinol, susceptible to contamination from oxygen and other airborne chemicals, the vacuum chamber offered by PBF-EB becomes a key advantage [44].

### 2.3. Directed Energy Deposition—Laser Beam 

Directed energy deposition with laser beam (DED-LB 3D) printing builds objects layer-by-layer using high-power lasers (3–4 kW) and powdered metal material. The powder is fed directly into the print head, where it is injected into a molten pool generated by the laser. Upon the solidification of each layer, the head moves on to deposit the next layer (Figure 5). This process typically occurs within a hermetically sealed chamber filled with argon gas to maintain a low-oxygen and low-humidity environment, thus minimizing oxide impurities [45].

While DED-LB offers advantages, it also comes with certain drawbacks. Notably, rapid solidification can lead to residual stresses within the printed object, necessitating subsequent heat treatment for stress relief. Additionally, the surface quality tends to be less smooth compared to other AM techniques. Finally, the high energy requirements and complex setup contribute to the relatively high cost and slower printing speeds of this method [19,45].

### 2.4. Directed Energy Deposition—Electron Beam—Wire 

Directed energy deposition with electron beam and wire (DED-EB-W) printing, also known as Electron Beam Additive Manufacturing (EBAM), leverages the power of electron beams and wire materials for 3D printing under a high vacuum environment. This method relies on a nozzle to directly feed wire into a molten pool generated by the electron beam, gradually building the desired shape layer-by-layer (Figure 6).

EBAM’s unique feature lies in its versatility: it allows printing using either a single wire (for alloys) or a combination of two separate wires (pure materials) that are mixed during the melting process by the electron beam. Both approaches offer specific advantages and drawbacks depending on the chosen materials and the optimized process parameters [46,47,48,49,50,51,52].

### 2.5. Directed Energy Deposition—Arc 

Directed energy deposition with arc (DED-arc), also known as wire arc additive manufacturing (WAAM), builds upon the DED principle and employs standard arc welding techniques—Tungsten Inert Gas (TIG) or Metal Inert Gas/Metal Active Gas (MIG/MAG)—to melt and deposit wire-based material. This method involves a unique approach: constructing the object by welding individual “caterpillars” (deposited material lines) side-by-side and layer-by-layer (Figure 7). Similar to other AM techniques, each layer is built by “drawing” the desired shape. However, DED-arc offers the flexibility to create components with either constant wall thickness (caterpillars) or variable thickness depending on the arc source control and process parameters [45,53].

As highlighted previously, various additive manufacturing (AM) techniques have unique production input parameters, such as scanning speed, layer thickness, and machine power. Optimizing and testing these parameters opens up a vast space for exploring different combinations of mechanical, microstructural, physical, and functional properties. Ultimately, the selection of the ideal property combination hinges on the intended application. Table 1 below summarizes the process parameters typically employed in NiTi alloy AM production.
materials-17-01248-t001_Table 1Table 1Process parameters of AM methods.ReferenceSystemScanning SpeedLayer ThicknessHatch SpacingLaser PowerSamples**Laser powder bed fusion**(Cai, 2023)[54]SLM EOSINT M280 (EOS GmbH, Krailing, Germany), 400 W Yb fiber laser, spot diameter of 80 μm800–1400 mm/s30 μm80 μm180 WDog bone-shaped, extracted from 8 × 8 × 8 mm^3^ blocks(Ge, 2023) [55]EOS M290 LPBF (EOS GmbH, Krailing, Germany)700–1100 mm/s30 μm100 μm150–300 WDog bone-shaped, extracted from 15 × 15 × 15 mm^3^ blocks(Chekotu, 2023)[56]Aconity MINI (Aconity 3D GmbH, Aachen, Germany),IPG Photonics Nd:YAG fiber laser system, laser power of 200 W and wavelength of 1068 nm600–1200 mm/s40 μm40–70 μm120–180 WExtracted from horizontally and vertically build blocks(Jiang, 2023)[57]RenAM 500E (Renishaw, Wotton-under-Edge, United Kingdom)500–1200 mm/s30 µm80 µm140 WLength 60 mm, width 15 mm, thickness from 0.15 mm to 1 mm(Liu, 2023) [58] AmPro SP100 SLM machine (AmPro Innovations, Notting Hill, Australia)900 mm/s30 μm60 μm190 WCylinders, 7 mm in diameter and 60 mm high (Sequeda Leon, 2023) [59] Renishaw AM400 system (Renishaw, Wotton-under-Edge, United Kingdom), with a 400 W ytterbium fiber laser, wavelength 1.070 μm, spot size 70 μm, laser energy density 44 and 85 J/mm^3^
40 µm64 µm81 and 156 W
(Zhan, 2023)[60]Concept Laser Mlab-R 3D printer (GE Additive, West Chester, OH, United States)250–800 mm/s25 µm110 µm50–95 WDog bone-shaped, tested cross-section5 × 2 mm^2^(Zhan, 2023) [41]Concept Laser Mlab-R 3D printer (GE Additive, West Chester, OH, United States)250–440 mm/s25 µm110 µm15–90 Wrepetitive scanning strategy
(Ehsan Saghaian, 2022) [34] Phenix PXM (3D Systems, Rock Hill, SC, United States), with a 300 W Ytterbium fiber laser with 80 µm focused beam125 mm/s30 µm40–240 µm50 and 100 WCylinders, 4.5 mm in diameter and 10 mm high(Obeidi, 2022) [61] Aconity MINI PBF-LB metal printer (Aconity 3D GmbH, Aachen, Germany), with a 200 W fiber laser of 1068 nm wavelength, spot size 50 µm400, 550, and 700 mm/s40 and 80 µm70 µm140, 160, and 180 W5 × 5 × 5 mm^3^(Zhang, 2022) [62] ProX 200 (3D Systems, Rock Hill, SC, United States) commercial PBF-LB system, Ytterbium fiber laser with wavelength 1070 nm, beam diameter 80 μm, and 300 W maximum power, laser energy density 60.92 J/mm^3^1080 mm/s38 µm80 µm200 W10 × 10 × 10 mm^3^(Nematollahi, 2021) [63] LPBF machine Prox 200 (3D Systems, Rock Hill, SC, United States)1250 mm/s
80 µm250 WTest-specific shape and dimensions(Obeidi, 2021) [64] Aconity MINI (Aconity 3D GmbH, Aachen, Germany) metal 3D printer, with a 200 W fiber laser manufactured by IPG Photonics with a wavelength of 1068 nm, spot size 40, 60, and 80 µm, laser energy density 11–100 J/mm^3^500, 1000, and 1500 mm/s90 µm40, 60, and 80 µm120, 150, and 180 W3 × 3 × 4 mm^3^(Wen, 2021) [65] HKM-125 (Wuhan Huake 3D Technology Co., Ltd., Wuhan, China), spot size 75 µm1000 mm/s30 μm
180 W10 × 10 × 5 mm^3^(Wang, 2019) [66] PBF-LB 250 HL machine (PBF-LB Solutions Group AG, Lübeck, Germany), spot size 80 µm200–1000 mm/s50 µm100 µm150 W10 × 10 × 2 mm^3^(Xiong, 2019) [20] Eplus M100-T (SHINING 3D, Hangzhou, China), with a maximum 200 W Yb-fiber laser of 50 μm in diameter500 mm/s30 µm80 µm120 W80 × 2.8 × 0.8 mm^3^(Shayesteh Moghaddam, 2018) [67] PBF-LB machine PXM (3D Systems, Rock Hill, SC, United States) with a 300 W Ytterbium fiber laser, laser energy density 55.5 J/mm^3^1250 mm/s30 µm120 µm250 WDog-bone shaped(Hamilton, 2017) [68] Type PXM PBF-LB commercial workstation (3DSystems, Rock Hill, SC, United States), beam diameter 0.08 mm, laser energy density 56 J/mm^3^1250 mm/s30 µm120 µm250 W4 × 4 × 8 mm^3^(Dadbakhsh, 2016) [69] An in-house PBF-LB machine (300 W Yb:YAG fiber laser with a beam diameter of about 80 µm1100 mm/s30 μm60 μm250 W6 × 6 × 12 mm^3^(Saedi, 2016) [70] PBF-LB machine PXM (3D Systems, Rock Hill, SC, United States), which is equipped with a 300 W Ytterbium fiber laser, beam diameter of 80 μm1250 mm/s30 μm120 μm250 WCylinders, 8 mm in diameter and 12 mm high(Dadbakhsh, 2015) [71] An in-house PBF-LB machine (continuous 300 W Yb:YAG fiber laser with a beam diameter of about 80 μm and a galvano scanner), laser energy density 111 and 126 J/mm^3^160 and 1100 mm/s30 µm75 and 60 µm40 and 250 WOctahedron porous structures(Bormann, 2014) [27]PBF-LB-Realizer 100 (PBF-LB-Solutions, Lübeck, Germany)107 to 297 mm/s50 μm120 μm56 to 100 WCylinders, 7 mm in diameter and 14 mm high(Zhang, 2013) [72]MCP Realize machine (MCP 250 HEK Tooling GmbH, Germany), Nd:YAG laser device with a wavelength of 1064 nm50–400 mm/s50 μm
100 W5 × 15 × 5 mm^3^(Bormann, 2012) [27]PBF-LB-Realizer 100 (PBF-LB-Solutions, Lübeck, Germany)133 to 171 mm/s50 μm120 μm60 to 80 WCylinders, 5 and 7 mm in diameter and 15 mm high(Shishkovsky, 2012) [73]YLR-50, continuous-wave Ytterbium fiber laser by IPG Photonics, spot size 60 µm100 and 160 mm/s60 µm100 µm50 Wrectangular(base 10 × 10 mm^2^)**Laser-directed energy deposition**(Wang, 2019) [66]LaserTec 65system (DMG MORI, Bielefeld, Germany), fiber-coupled diode laser (Laserline GmbH, Mülheim-Kärlich, Germany)16.67 mm/s1000 µm1500 µm500 and 1000 W20 × 20 × 15 mm^3^(Hamilton, 2017) [68]Direct Digital Deposition (CIMP-3D, State College, PA, United States) in a custom-built atmospherically controlled chamber, beam diameter 4 mm, laser energy density 61 J/mm^3^10.6 mm/s700 µm2200 µm1000 WMicromachined from a build volume of 56 × 13 × 10 mm^3^(Halani, 2013) [74]Lens 750™ system (Optomec, Albuquerque, NM, United States) with a continuous fiber laser (wavelength: 1.064 μm), spot size 0.5 mm12.5 mm/s250 µm500 µm250 WCylinders, 10 mm in diameter and 25.4 mm high(Malukhin, 2006) [75]Nd:YAG laser DMD machine with a feedback control system (courtesy of Dr. R. Kovacevic at SMU)10.16 mm/s381 µmm
250–600 WCylinders, 25.4 mm in diameter and 2.54 mm high**Selective electron beam melting**(Fink, 2023)[76]PBF-EB machine Freemelt One (Freemelt Holding, Mölndal, Sweden)
75 μm200 μm
10 × 10 × 15 mm^3^(Wang, 2019) [66]Arcam EBM A2X system (GE Additive, West Chester, OH, United States)



20 × 20 × 15 mm^3^

Across the majority of investigated studies, NiTi alloy samples were fabricated using powder materials. Various factors influence the resulting properties, including the material’s chemical composition, particle size, manufacturer, and fabrication method. A summary of these parameters and their potential impacts is presented in Table 2 below.
materials-17-01248-t002_Table 2Table 2Powder properties.ReferencePowder CompositionPowder Particle Size (µm)Powder ProducerPowder Type/ Fabrication MethodNiTi**Laser powder bed fusion**(Cai, 2023)[54]50.7 at.%49.3 at.%19–53,D50 = 33AMC Powders Co. Ltd., Beijing, ChinaElectrode induction-melting gas atomized(Ge, 2023)[55]Nearly equal atomic ratioD10 = 16.5, D50 = 31.2, D90 = 48
Pre-alloyed aerosolized NiTi powder(Chekotu, 2023)[56]49.9 at.%50.1 at.%D10 = 12.3, D50 = 28.1, D90 = 57.3Ingpuls GmbH, Bochum, GermanyElectron induction melting gas atomized(Jiang, 2023)[57]50.6 at.%49.4 at.%D10 = 21.9, D50 = 35.1, D90 = 55.6Minatech Co., Ltd. Shenzhen, ChinaAlloy powder(Zhan, 2023)[60]55.98 wt.%44.02 wt.%10–60, D50 = 37Advanced Materials Technology, Co., Ltd.Pre-alloyed NiTi powder(Zhan, 2023) [41]50.91 at.%49.09 at.%15–53, D50 = 37Advanced Materials Technology, Co., Ltd.Pre-alloyed NiTi powder(Sequeda Leon, 2023) [59]50.47 at.%49.53 at.%15–53Hermus & OEMSichuan Hermus Industry Co., Ltd, Chengdu, China–(Ehsan Saghaian, 2022) [34]50.8 at.%49.2 at.%Mean size 50TLS Technik GmbH (Bitterfield, Germany)Electrode induction-melting gas atomized from NiTi ingot(Obeidi, 2022) [61]50 at.%50 at.%D10 = 16.7, D50 = 31.5, D90 = 56.6Ingpuls GmbH, Bochum, GermanyGas atomized Nitinol powder(Zhang, 2022) [62]50.8 at.%49.2 at.%10–60; D50 = 29, D80 = 38Carpenter Technology Corp, Philadelphia, PA, United StatesElectrode induction-melting gas atomized(Nematollahi, 2021) [63]50.8 at.%49.2 at.%16–63–Pre-alloyed powder, gas atomized(Obeidi, 2021) [64]50.10 at.%49.99 at.%D10 = 16.7, D50 = 31.5,D90 = 56.6Ingpuls GmbH, Bochum GermanyGas atomized(Wen, 2021) [65]50.73 at.%49.27 at.%D10 = 20.6, D50 = 34.0, D90 = 53.1AMC Powders Co. Ltd., Beijing, ChinaGas atomized50.93 at.%49.07 at.%D10 = 22.1, D50 = 36.0, D90 = 58.5AMC Powders, ChinaGas atomized51.27 at.%48.73 at.%D10 = 34.4, D50 = 48.5, D90 = 67.9AMC Powders, ChinaGas atomized(Xiong, 2019) [20]50.4 at.%49.6 at.%15–53–Gas atomized(Shayesteh Moghaddam, 2018) [67]50.1 at.%49.9 at.%25–75TLS Technik GmbH (Bitterfield, Germany)Electrode induction-melting gas atomized(Hamilton, 2017) [68]50.09 at.%49.91 at.%25–75–Pre-alloyed(Dadbakhsh, 2016) [69]50.2 at.%49.8 at.%25–45Raymor Industries Inc., Boisbriand, QC, CanadaPlasma atomized NiTi(Saedi, 2016) [70]50.8 at.%49.2 at.%25–75TLS Technik GmbH (Bitterfield, Germany)Electrode induction-melting gas atomized from NiTi ingot(Dadbakhsh, 2015) [71]55.2 wt.%44.8 wt.%25–45Raymor Industries Inc., CanadaPlasma-atomized NiTi powder(Bormann, 2014) [77]55.96 wt.% (+0.05 wt.% O, C, H)43.99 wt.%35–180Memry GmbH, Weil am Rhein, GermanyPre-alloyed(Zhang, 2013) [72]55 wt.%45 wt.%Ni D50 = 30, Ti D50 = 35Powders by Sulzer Metco Technologies Co. Ltd., Winterthur, SwitzerlandA mixture of Ni and Ti powders(Bormann, 2012) [27]56.1 wt.%43.9 wt.%D50 = 60MEMRY GmbH, Weil am Rhein, GermanyPre-alloyed gas atomized(Shishkovsky, 2012) [73]55 wt.%45 wt.%D10 = 11.4, D50 = 25.5, D 90 = 37.5TLS Technik GmbH&Co (Niedernberg, Germany)Pre-alloyed gas atomized**Laser-directed energy deposition**(Hamilton, 2017) [68]53 wt.% (47.9 at.%)47 wt.% (52.1 at.%)Ni D50 = 101.7, Ti D50 = 94.1–Elementally blended Ni and Ti powders(Halani, 2013) [74]57 at.%43 at.%–Atlantic Equipment Engineers, Upper Saddle River, NJ, United StatesA mixture of Ni and Ti powders(Malukhin, 2006) [75]55.5–56 wt.%44–44.5 wt.%10–180UDIMET NITINOLPre-alloyed**Selective electron beam melting**(Fink, 2023) [76]48.9 at.%51.1 at.%D10 = 45, D50 = 64.4, D 90 = 89.1Eckart TLS GmbH (Bitterfeld-Wolfen, Germany)Argon-atomized pre-alloyed Ti-rich NiTi powder

## 3. Functional Properties of NiTi Alloys

The unique properties of alloys with the shape memory effect (SME) are due to the reversible martensitic transformation when the high-temperature phase (austenite) changes to a low-temperature phase (martensite) and vice-versa. The phase transformation is accompanied by a change in crystallographic structure (Figure 8), e.g., a body-centered cubic structure (B2) of the austenite (Figure 8a) spontaneously transforms into more complicated crystal structures of different variants of the martensite (Figure 8b) [5]. The recovery of the original shape after deformation (Figure 8c and curve AB in Figure 9) passes when thermal treatment is applied. These processes depend on four basic phase transformation temperatures—austenite start *A_s_*, austenite finish *A_f_*, martensite start *M_s_*, and martensite finish *M_f_* [5] (curve DA in Figure 9). 

An increasing temperature (curve CD in Figure 9) above the *A_s_* induces martensite transition in austenite and the recovery of the original geometrical shape simultaneously. This process is completed above *A_f_* when the NiTi alloy is fully in an austenite state (Figure 8 and Figure 9). At cooling below *M_s_*, the austenite transitions to twinned martensite while keeping its shape. The martensite transformation is completed below *M_f_* [5,78]. So, the evolution of the shape variation upon deformation and successive heating processes is related to the existence of the thermoelastic martensite. 

Another important effect of the NiTi alloy applied in different areas of industry or medicine is the so-called superelasticity. At mechanical loading within a specific temperature range (below *M_d_* and above *A_f_*) [5,13], the mechanically stressed austenite transforms into stress-induced martensite (SIM) (curve E in Figure 9). When unloading, the SIM transforms back to austenite, recovering the original shape. The temperature *M_d_* gives the highest value at which the SIM can be induced by applied stress. At all temperatures above *M_d_*, loading the NiTi alloy leads to permanent deformation, as for a common metallic material. The superelasticity can result in a large elastic response [5,79,80]. Depending on the structure and the thermomechanical treatment, NiTi alloys can offer an extensive reversible strain of 6–8% [5,79,80].

The transformation temperatures, their ranges, and hysteresis significantly affect the final properties and following applications of the NiTi products. As it was proved by several authors, the transformation temperatures are significantly dependent on the composition ratio of Ni/Ti and the purity of the alloys [5,15]. Indeed, by changing the Ni content by 0.1 at.%, the *M_s_* temperature varies by 10–15 °C [5]. The shape memory and superelasticity are strongly influenced by the content of impurities. The highest impact is displayed by oxygen and carbon, which can deplete the NiTi matrix of Ti atoms by forming oxide or carbon inclusions and shifting transformation temperatures. It was stated that the shape memory effect at room temperature manifests in Ti-rich NiTi alloys; conversely, superelasticity is displayed in Ni-rich NiTi alloys [5].

**Figure 8 materials-17-01248-f008:**
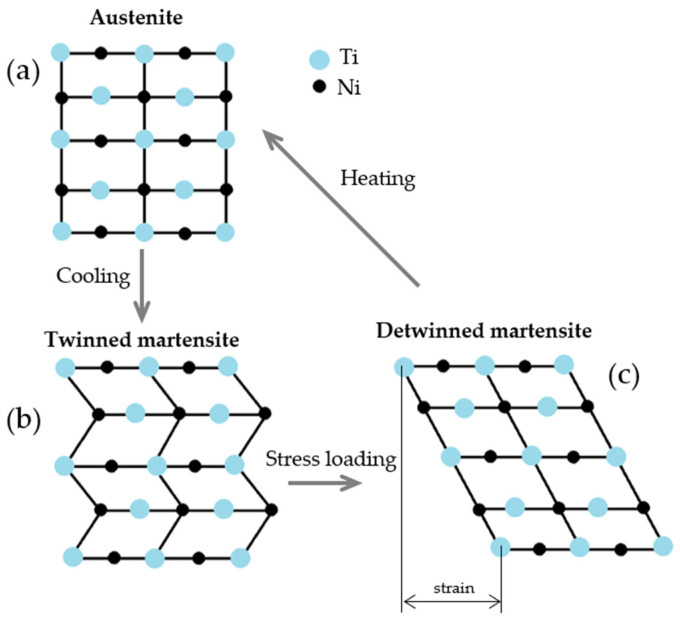
Phase transformation at temperature changes producing shape memory effect: (**a**) austenite, (**b**) twinned martensite, and (**c**) deformed or detwinned martensite.

**Figure 9 materials-17-01248-f009:**
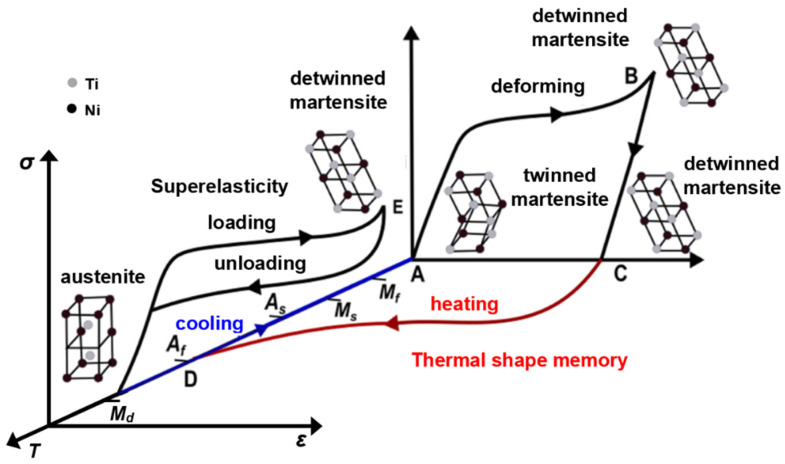
Development of stress–strain–temperature dependence for Nitinol at superelastic or thermoelastic response [81].

As mentioned above, any deviation of the binary NiTi alloy composition from 50:50 stoichiometry develops the transformation temperature shifts, and this can be further evoked through precipitation of several types of precipitates formed in alloy microstructure after primary processing, for example, NiTi_2_ or TiO_2_ and TiC in the case of contamination [5,15,82,83,84]. Furthermore, thermal treatment of the NiTi alloy subsequent to casting or additive manufacturing induces diffusion processes and additional precipitation of various phase particles: metastable Ni_4_Ti_3_, metastable Ni_3_Ti_2_, or stable Ni_3_Ti [85,86,87,88,89,90,91,92,93,94]. The heat treatment, also called post-processing treatment [15], can consist of annealing (for residual stress release, homogenization, or solution treatment) or additional aging (precipitation strengthening) [82,90,91,92,95]. The parameters, e.g., temperature and duration, of thermal treatment depend on the Ni and Ti contents as well as on the intended application. Usually, temperatures of annealing and aging range from 800 to 1050 °C and from 300 to 600 °C for annealing and aging, respectively [96,97,98,99].

## 4. Evaluating Properties of AM NiTi Alloy 

The chemical composition of the NiTi alloy, processing parameters, and thermal or thermomechanical treatment significantly influence the microstructural changes and final properties of the products [100,101,102]. These properties can be assessed through various physical, chemical, and mechanical methods, including microscopy techniques, thermal analysis, X-ray analysis, resistivity measurement, and mechanical testing [103,104,105].

Different microstructural and structural features, such as grain size and shape, the distribution of grains or precipitates, porosity, unmolten particles, internal cracks, and the ratio of martensite–austenite phases, can be observed or detected using various microscopy techniques [97]. Optical microscopy, scanning electron microscopy (SEM), and transmission electron microscopy (TEM) offer diverse levels of detail [96,106,107]. For instance, analyzing the quality of the feedstock material, like the alloy powder, before additive manufacturing (AM) is crucial for ensuring the final properties of Nitinol [100,101,102]. SEM, energy-dispersive X-ray spectroscopy (EDS), X-ray computed tomography (XCT), laser diffraction, and Hausner ratio measurements can control key features such as particle shape, size, composition, and flowability [101,102].

The macrostructure or microstructure of AM products, depending on the melting and solidification of the melt pool, is investigated after metallographic preparation consisting of grinding, polishing, and potential etching. Etching the polished surface usually reveals martensite–austenite phases or grain size. Common etching solutions include HF:HNO_3_:CH_3_COOH (3:5:3 ratio) for phases or HF:HNO_3_:H_2_O (2:5:10 ratio) for grain size [103,104,105]. SEM analysis provides detailed observations of topographic features, small particles, fracture surfaces, pores, and Ni_4_Ti_3_ precipitates, offering deeper insights compared to optical microscopy. Structures on a very fine scale are possible to detect only by means of TEM analysis [96,97]. This technique requires preparing thin foils from the material, typically achieved through mechanical grinding followed by twin-jet electropolishing in a CH_3_COOH:HClO_4_ (95:5 in vol.%) electrolyte solution [106].

Differential scanning calorimetry (DSC) or differential thermal analysis (DTA) are employed to investigate physical properties, particularly phase transition temperatures in both the input powder and final printed or post-processed products [30,104,106,108,109]. When applying a constant heating or cooling rate within specific temperature ranges encompassing potential martensite or austenite phases, exothermic and endothermic peaks in DSC curves indicate phase transitions in small samples [30,104,106,108,109]. Compared to the alloy powder, the transformation temperatures *A_f_* and *M_s_* in the final printed product can differ due to Ni loss, inclusion content, and various precipitates formed during the manufacturing process [108]. Higher Ni loss is associated with more pronounced shifts of *A_f_* and *M_s_* to lower and higher temperatures, respectively. Additionally, manufacturing parameters like hatch spacing, scanning speed, and laser power can influence the position and sharpness of transformation peaks [109].

The chemical composition of the matrix and other phases are investigated by energy dispersive X-ray spectroscopy (EDX), usually in combination with SEM analysis or secondary ion mass spectrometry (SIMS) [73,75]. The SIMS method is most often used to analyze the exact chemical composition of the material surface and possible contamination with undesirable substances and can detect elements, isotopes, or molecules with high sensitivity. On the other hand, the accuracy of the EDX analysis, which allows for detecting relatively abundant elements in the alloy, is rather lower, and some authors recommend using a more sensitive method to detect the contamination of the alloy [73,75].

For the identification of the crystallographic structure of the various transformed or precipitated phases in the NiTi alloy powder or printed products, X-ray diffraction (XRD) analysis was successfully used [66,110,111]. Large variations of processing parameters of additive manufacturing were studied in [30,66,109,112]. The peaks of XRD revealed the presence of both B2 austenite and B19′ martensite in as-built Nitinol samples with 55.64 weight % of Ni [109]. Increasing the intensity of the B19′ phase reflections in X-ray diffraction patterns proved to increase the volume fraction of the B19′ phase with decreased scanning speed or hatch spacing and increased laser power. 

By simple tests that respect temperature dependence and transformation temperatures, the shape memory behavior can be verified on the additively manufactured samples under investigation. The figures below (Figure 10 and Figure 11) clearly show both temperature and stress hysteresis, i.e., the force required for loading is greater than the force for unloading. The area of the hysteresis loop (the area between the loading and unloading curves) is the dissipated heat energy that is used in damping vibration and shock. The area under the unloading curve is the stored energy per unit volume of material for unloading, and the area under the loading curve is the total deformation energy of the material. The efficiency of energy storage is given by the ratio of stored energy to total energy [104]. 

Mechanical testing offers invaluable insights into the remarkable properties of NiTi alloys, like shape memory, superelasticity, and pseudoplasticity. These properties, alongside crucial material characteristics like modulus of elasticity, tensile/compressive strengths, and hardness, significantly impact the alloy’s suitability for various applications [110]. Static tensile and compression tests, typically conducted at room temperature (below the *M_f_* temperature), reveal highly asymmetric stress–strain curves between the two loading modes. Simple tensile tests can quantify tensile strength, maximum strain, and elastic modulus (derived from the linear region’s slope in the stress–strain curve). Compression tests provide similar data. Analyzing an AM product comprehensively requires accounting for potential anisotropy, necessitating testing under both tension and compression and potentially bending. However, the literature review identified a notable gap regarding bending properties despite their relevance in the human body, where bending often dominates loading conditions.

Hardness constitutes another critical parameter assessed in AM NiTi materials. Various methods measure hardness by applying a defined indenter to the material and gauging the resistance based on the resulting impression. The Vickers method is typically employed for Nitinol. The hardness of the material used for implantology is important for several reasons, such as biocompatibility, mechanical strength, wear resistance, and surgical manipulation. The effect of input laser power (Figure 12) and Nitinol composition (Figure 13) on hardness is shown in the figures below.

## 5. Effect of AM Technologies on the Properties

### 5.1. Effect of PBF-LB Method 

Energy density plays a crucial role in shaping the properties of additively manufactured (AM) NiTi alloys via Powder Bed Fusion-Laser Beam (PBF-LB) methods. As demonstrated by Wang et al. [66], insufficient fusion (low energy density, 40 J/mm^3^) leads to defects, while intermediate energy density (80 J/mm^3^) yields better fusion but inhomogeneous microstructure. A higher energy density (130 J/mm^3^, associated with slower scanning speeds) promotes larger melt pools and better mixing, but excessive laser power can create keyhole defects [66]. This emphasizes the importance of optimizing energy density to achieve full parts with minimal microstructural inhomogeneity, as confirmed by [20,77,113]. 

Martensitic transformation temperatures also exhibit dependence on energy density. Studies by [7,71] revealed that higher energy density increases these temperatures, while low scanning speed and laser power favor pseudoplastic behavior. However, balancing power and speed is crucial to avoid geometric and surface inaccuracies [71]. Optimizing bulk energy density ensures proper solidification and evaporation control [34]. Alternatively, pulsed laser modes offer significant energy reduction and better temperature control [61].

Obeidi et al. [61] highlight the importance of meticulous parameter selection on single-line prints to save time and cost while gaining valuable insights into melt properties. Larger hatch spacing and lower laser power increase porosity, while smaller spacing and higher energy density enhance transformation temperatures and recovery ratios [34,59]. Laser frequency impacts surface roughness, and layer thickness affects density and compressive strength [34].

Build orientation and scanning strategy significantly influence mechanical and shape memory properties, as shown by Moghaddam et al. [67]. Horizontal printing with alternating x/y scanning yielded the highest tensile strength (606 MPa) and elongation (6.8%), with comparable strain recovery, while ±45° scanning deteriorated these properties [63,67]. Loading direction based on deposition also plays a key role, with XY specimens loaded perpendicular to layers exhibiting superior tensile properties compared to ZX specimens loaded parallel [38,63,67].

Figure 13 shows that the hardness of the sample depends not only on the composition of the NiTi alloy but also on the AM process parameters. PBF-LB produces parts with higher microhardness (540–734 HV) than cast samples, attributed to matrix hardening under rapid laser cooling [73]. Nickel content in the NiTi alloy significantly impacts superelasticity and shape memory. PBF-LB processes often involve high temperatures, leading to preferential nickel evaporation due to its lower boiling point, resulting in higher titanium content and increased transformation temperatures [5]. Additionally, high-temperature processing can lead to the precipitation of nickel-rich secondary phases, further depleting the matrix of nickel [7,59].

Wen et al. [65] demonstrated that lower Ni content (50.73–51.27 at.%) promotes better shape memory, while higher Ni content (50.93–51.27 at.%) favors superelasticity under the same PBF-LB conditions. It was also found that Ni evaporation during production ranges from 1.6 to 3.0 wt.% depending on the initial composition and melting cycles [36]. Interestingly, increasing melt cycles not only increase Ni evaporation but also decrease defects and improve alloy homogeneity [36]. This highlights the potential for tailoring final properties by adjusting the initial composition.

Heat treatment is commonly employed to lower transformation temperatures [7,25,26,27,28,29,30,31]. Processes like solution annealing and aging dissolve secondary phases and restore superelastic behavior [22,23,34,59].

The overview of the effect of processing parameters (Table 1) on the mechanical properties of materials produced by PBF-LB (Table 3) is summarized in Figure 14. There is an obvious dependence between the laser power and the scanning speed: the higher scanning speed requires higher laser power to melt powder (Pearson correlation coefficient 0.86, *p* < 0.001). Both the laser powder and the scanning speed correlate with the ultimate tensile strength (Pearson correlation coefficient 0.75 and 0.65, respectively, *p* < 0.001) but not with elongation (Pearson correlation coefficient 0.29 and 0.35; *p* = 0.11 and *p* = 0.05, respectively), Figure 15. However, there is no significant correlation between the laser energy density and both the ultimate tensile strength and the elongation (Pearson correlation coefficient −0.25 and 0.03; *p* = 0.16 and *p* = 0.88, respectively).

In conclusion, the PBF-LB method offers enormous potential for fabricating NiTi alloys with tailored properties for diverse applications. However, careful optimization of energy density, laser parameters, Ni content, build orientation, and heat treatment is crucial to achieve the desired outcome. 

### 5.2. Effect of PBF-EB Method

Chamber temperature plays a crucial role in the success of PBF-EB (electron beam powder bed) fabrication of NiTi alloys. As demonstrated by Wang et al. [66], preheating temperatures significantly impact the process. While the highest temperature (600 °C) resulted in powder ignition and chamber pressure increase, the lowest temperature (350 °C) led to insufficient bonding and porosity. The chosen intermediate temperature (450 °C) avoided ignition but still faced exothermic reactions due to the mixture of elemental Ni and Ti powders [66,76].

This challenge is overcome by using pre-alloyed NiTi powder instead, as shown by Fink et al. [76]. Their approach enabled fully dense samples at 820 °C with lower bulk energy density and build speed, eliminating the exothermic reaction associated with elemental powder mixtures. The resulting microstructure exhibited characteristic columnar grains aligned with the layer orientation [76].

Mechanical properties of PBF-EB NiTi vary across studies. Fink et al. [76] achieved a tensile strength of 736 MPa and 5.2% elongation, while Hayat et al. [115] reported higher values (1411 MPa and 11% elongation). However, some studies show lower properties compared to reference NiTi materials, often with predominantly brittle fracture [115].

These contrasting outcomes highlight the sensitivity of mechanical properties to chamber temperature, powder selection, and process parameters. PBF-EB offers higher energy density and scanning speed, potentially leading to higher transformation temperatures compared to PBF-LB [7]. However, the presence of a vacuum system makes this method less cost-effective in comparison to PBF-LB [19].

### 5.3. Effect of DED-LB Method

Similar to other laser-based 3D printing methods, scanning speed in DED-LB significantly influences the NiTi microstructure. Lower speeds promote axial grains, while higher speeds yield columnar ones [116]. The deposition and melting rate of the laser beam also affect the final product’s properties. Much like PBF-LB, DED-LB prints may contain unwanted secondary phases, necessitating further heat treatment to enhance NiTi’s shape memory and superelasticity, particularly susceptible to thermal effects during AM [116]. Wang et al. [66] investigated laser powers of 500 W and 1000 W, finding better fusion at 1000 W. However, other studies used diverse parameters (Table 1).

An undesirable hard Ti_2_Ni phase was observed in the NiTi phase, contrasting the pure martensitic NiTi and causing brittleness evidenced by low ductility [66]. This relates to Ni depletion due to its faster evaporation at high temperatures and potentially uneven powder separation in the nozzle.

Another study [117] reported microstructural anisotropy due to Ni_4_Ti_3_ precipitates but without nickel depletion due to high initial powder content. These precipitates, typically appearing only after post-aging heat treatment [74], raise the critical stress for permanent deformation (slip) and, thus, enhance superelasticity.

Subsequent heat treatment can dissolve the Ti_2_Ni phase and improve homogeneity, although limited by nickel content [66,74,75]. The strength and ductility of DED-LB samples vary across studies due to diverse process parameters and heat treatments. For example, [66] reported specimens with brittle fracture and low tensile strength (250 MPa), narrow hysteresis loops under cyclic loading, and hardness indicating a hard, brittle phase. Conversely, heat-treated samples in Halani et al. [74] achieved excellent strength (>2.5 GPa), shape memory, and superelasticity (up to 4.2% strain recovery and 3% superelasticity). In conclusion, PBF-EB can successfully fabricate high-quality, nearly non-porous NiTi samples with retained shape memory and superelasticity. Tailoring mechanical properties and these effects can be achieved through various heat treatments (aging and quenching) [66,74,75,117]. Furthermore, combining different NiTi powder compositions, heat treatments, and their conditions allows programming the transformation temperatures and, thus, the desired functional properties [74].

### 5.4. Effect of DED-EB-W Method

Process parameters in DED-EB-W (or EBAM) significantly impact the quality and shape of NiTi parts. Excessive beam power causes material spatter, hindering deposition and layer control [46]. Optimizing wire feed speed is crucial, as insufficient speed leads to defects, while excessive speed is less detrimental.

Despite varying process parameters, all DED-EB-W-fabricated parts (using NiTi wire) exhibited superelastic and shape memory behavior, with transformation temperatures dependent on fabrication parameters, particularly beam power affecting nickel evaporation [46]. This allows for the “programming” of fabricated parts, with a high transformation temperature leading to good shape memory in the ground state and a low transformation temperature enabling superelasticity after heat treatment [46].

Expanding this concept, researchers utilized the dual-wire DED-EB-W method with separate Ni and Ti wires. The resulting microstructure displayed continuous coarse columnar grains and a dominant NiTi phase, indicating successful synthesis within the melt pool [47]. Both mechanical properties and martensitic phase transformation were impressive, with compressive strengths reaching 2.9 GPa and ground-state samples exhibiting up to 88% shape memory recovery [46,47]. Crucially, dual-wire EBAM allows for the independent programming of Ni and Ti wire feed rates, providing control over the final alloy composition and enabling the fabrication of “intelligent” parts with tailored properties [47].

### 5.5. Effect of DED-arc Method

DED-arc offers advantages for NiTi printing, including low defect rates, consistent wire composition, and a smaller surface area, leading to fewer impurities and a smaller reactive surface during heat treatment [118]. While not highly accurate, its efficiency and simplicity make it suitable for large, geometrically simple parts. It boasts higher deposition speeds, material efficiency, and lower costs compared to other DED processes [45,53]. Studies like Liu et al., 2022 [119] demonstrate the successful fabrication of homogeneous NiTi prints by optimizing source movement and wire feed speed. Additionally, interlayer temperature control plays a crucial role, with optimized temperatures around 200 °C leading to improved mechanical properties, microstructure, and reduced micropores [120]. This results in tightly bonded layers with no visible defects [119,121].

Mechanical properties vary based on wall height and processing parameters. Generally, microhardness, elongation, and critical stress increase with height [119,121]. Liu et al., 2022 [119] report achieving a hardness of 230–340 HV, elongation of 6.39%, and critical stress of 643 MPa [119,121], while Singh et al., 2023 report even higher local hardness (730 HV) and tensile strength (910 MPa) at 200 °C [120]. Reversible strain reached 3.06% after 10 cycles, with potential for improvement by optimizing the deposition rate [119]. Notably, property tailoring is possible through patterning direction, where increasing strength (up to 668.3 MPa) comes at the cost of reduced elongation (5.0% max) [122].

However, challenges remain. Resnina et al., 2021 [123] observed a brittle Ti_2_Ni phase due to NiTi phase decay at grain boundaries [123]. Compositional heterogeneity exists across layers, and achieving consistent mechanical properties throughout the part remains a hurdle [123]. Additionally, different layers can exhibit different phases due to varying interlayer transition temperatures, impacting superelasticity and shape memory effect across the part [124].

DED-arc holds significant promise for NiTi printing due to its advantages and ability to tailor properties. However, addressing microstructural and compositional inconsistencies, along with optimizing interlayer temperatures, is crucial for achieving fully uniform and reliable functional properties in these parts.

Table 3 summarizes mechanical properties in tension and compression across several additive manufacturing methods at various conditions. The authors monitored the changes in these properties at differing source power, scanning speed, building direction, or porosity. In addition, some sources also reported hardness.

**Table 3 materials-17-01248-t003:** Mechanical properties in static loading.

Reference	Method/ Specification		Ultimate Tensile Strength (MPa)	Ultimate Tensile Strain (%)	Ultimate Compressive Strength (MPa)	Ultimate Compressive Strain (%)	Hardness
**Powder bed fusion—laser beam**							
(Cai, 2023) [54]	PBF-LB different scanning speed	800 mm/s	870	6.7			
		1000 mm/s	813	6.3			
		1400 mm/s	829	8.1			
(Ge, 2023) [55]	PBF-LB different loading direction	Parallel to building	757	8			
		Transverse to building	644	11			
(Chekotu, 2023) [56]	PBF-LB	Horizontal			360–420		290–454 HV
		Vertical			395–430		248–362 HV
(Jiang, 2023) [57]	PBF-LB different specimen thickness	0.15 mm	297	6.9			
		1 mm	446	9.7			
(Zhan, 2023) [60]	PBF-LB different laser power, scanning speed, and VED	63 W, 454 mm/s, 50.46 J/mm^3^	403	7.6			
		83 W, 724 mm/s, 41.69 J/mm^3^	573	10.1			
(Zhan, 2023) [41]	PBF-LB different laser power and scanning speed	45 W, 250 mm/s	469	6.2			
		60 W + 45 W, 250 mm/s	440	6.8			
		60 W, 440 mm/s	520	6.9			
(Ren, 2022) [35]	PBF-LB different laser power	60 W	607.6	5.62			
		90 W	769.0	5.45			
		120 W	824.2	5.19			
		150 W	683.1	4.54			
(Yu, 2022) [114]	PBF-LB different laser power	200 W	653	8.08			
		91 W	588	10.97			
(Obeidi, 2021) [64]	PBF-LB different VED	100 J/mm^3^			3400	32	
		83.33 J/mm^3^			3320	32.6	
		50 J/mm^3^			2800	30.8	
		11 J/mm^3^			1500	26.2	
(Xiong, 2019) [20]	PBF-LB		700	15.6			
(Saedi, 2018) [28]	PBF-LB						318 HV
(Shayesteh Moghaddam, 2018) [67]	PBF-LB horizontal		601	6.8			
	PBF-LB vertical		331	3.5			
	PBF-LB edge		482	5.1			
(Taheri Andani, 2017) [23]	PBF-LB different porosity	dense			1649	30.2	
		32%			1035	28.3	
		45%			728	24	
		58%			410	15.6	
**Directed energy deposition—laser beam**							
(Wang, 2019) [66]	DED-LB		250 (brittle)				422 ± 10.8 HV
**Powder bed fusion—electron beam**							
(Fink, 2023) [76]	PBF-EB		735	5.2			
(Zhou, 2019) [104]	PBF-EB		1411	11.8			
**Electron beam freeform fabrication**							
(Li, 2023) [125]	EBF^3^ different post-heat treatment	Solution	615.9	6.15			
		Aging	679.6	7.14			
(Li, 2021) [48]	EBF^3^	Building dir.	578.9	4.52			
		Horizontal dir.	599.5	6.09			
**Wire arc additive manufacturing**							
(Huang, 2024) [126]	WAAM				2214.3	31.7	260–531 HV
(Han, 2023) [127]	WAAM				2720	46.6	237–531 HV
(Teng, 2023) [128]	WAAM, lower, middle, and upper region	homogeneous	607–654	12.7–15.4	2745–2950	40.7	240–337 HV
		heterogeneous	556–740	13–22	2534–3473	38.6–43.9	235–303 HV
(Khismatullin, 2022) [129]	WAAM, lower, middle, and upper region	As-build	576–592	4.3–6.0			349–382 HV
		Heat-treated	467–561	6.7–7.0			330–348 HV

## 6. Conclusions

NiTi alloys hold potential due to their unique combination of superior properties, including superelasticity, reversible shape memory, low stiffness, biocompatibility, and excellent corrosion resistance. This review explores the exciting intersection of these alloys with additive manufacturing (AM), particularly focusing on the ability to manipulate shape memory behavior through processing parameters. Recent research demonstrates the remarkable success of AM in fabricating NiTi components. Notably, by strategically modifying scan parameters and strategy, the resulting microstructure and transformation temperatures can be precisely controlled, paving the way for engineering functional properties (shape memory or superelasticity) for specific applications. This level of customization was previously unattainable with traditional manufacturing methods.

However, it is crucial to emphasize that printability and functional properties considerably vary across methods, even when utilizing the same NiTi powder. Figure 16 shows that there is high variability within the method and between the methods as well. This underscores the critical need to optimize each AM process individually to achieve desired results. Process settings profoundly influence the properties of these additively manufactured materials, further highlighting the importance of tailored optimization. 

While AM provides immense design freedom, unlocking the creation of complex geometries like porous structures with defined unit cells, it is important to acknowledge that achieving consistent properties throughout these intricate designs remains a challenge. Despite these challenges, the potential to influence shape memory behavior through these designs offers exciting possibilities for innovative applications.

Overall, the combination of NiTi’s unique properties and AM’s advanced capabilities creates a promising avenue for developing materials with tailored functionalities. This emerging field holds immense potential for diverse applications across various sectors, including biomedical implants, microfluidics, and aerospace components. However, it is crucial to address the existing challenges related to process optimization and achieving consistent properties in complex geometries through further research and exploration to fully unlock the transformative potential of these methods.

## Figures and Tables

**Figure 1 materials-17-01248-f001:**
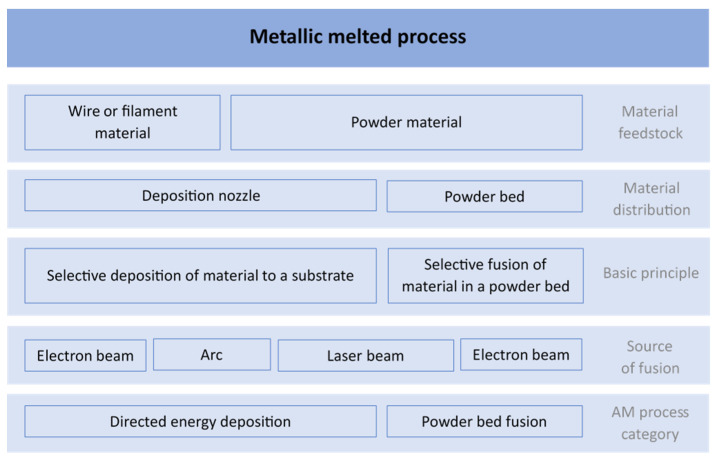
Overview of additive manufacturing methods for metallic materials [24].

**Figure 2 materials-17-01248-f002:**
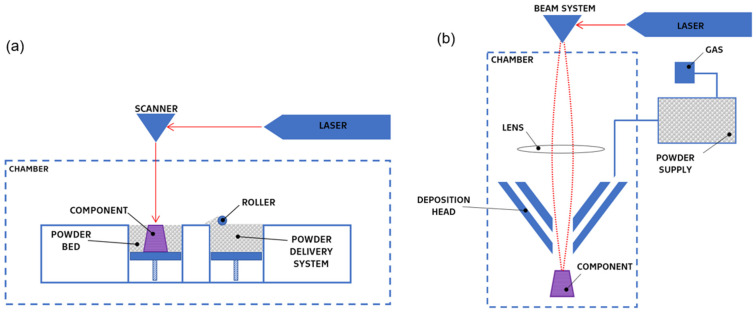
(**a**) Schema of powder bed system; (**b**) schema of directed energy deposition system with nozzle.

**Figure 3 materials-17-01248-f003:**
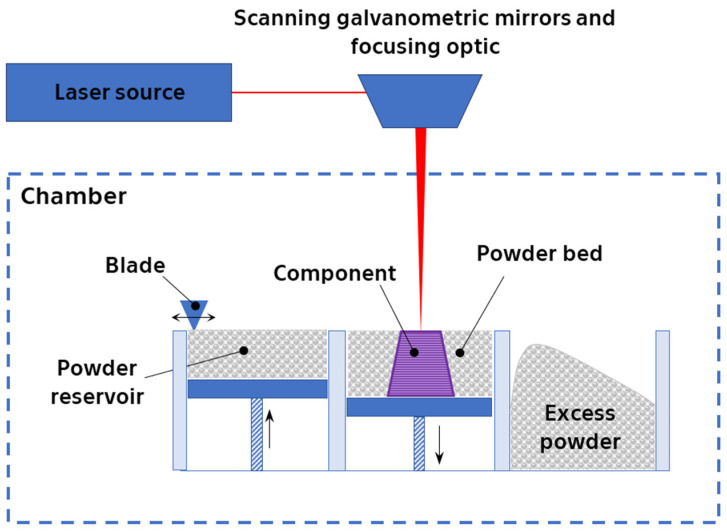
Schema of the laser powder bed fusion process (PBF-LB).

**Figure 4 materials-17-01248-f004:**
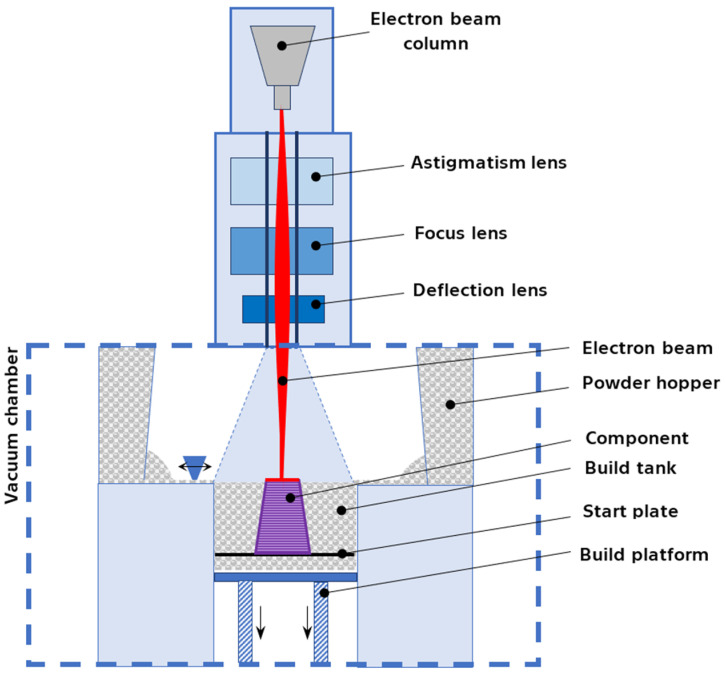
Schema of the electron beam melting process (PBF-EB).

**Figure 5 materials-17-01248-f005:**
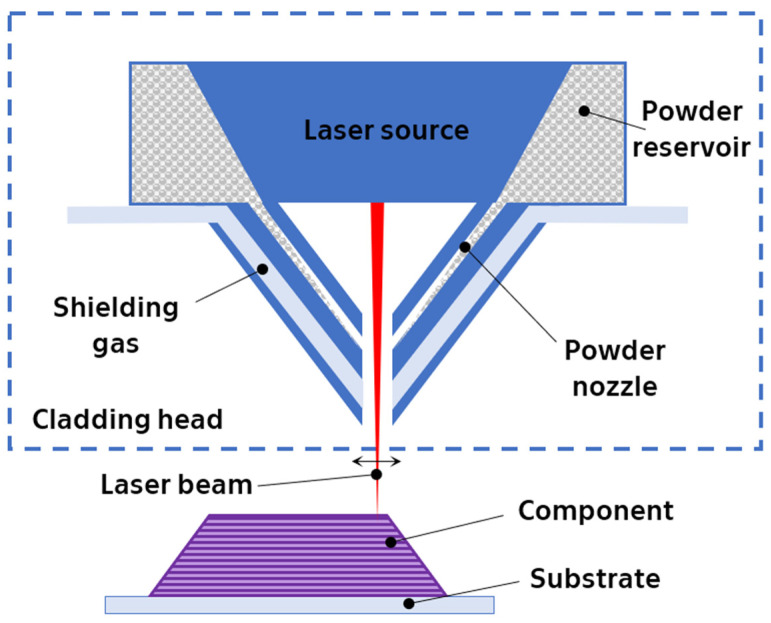
Schematic illustration of directed energy deposition additive manufacturing set-up and build coupon.

**Figure 6 materials-17-01248-f006:**
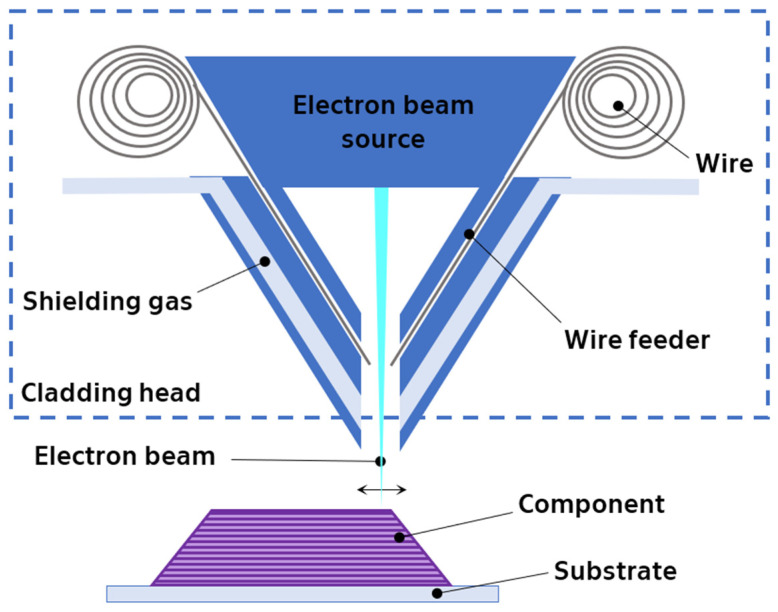
Schematic diagram of electron beam wire melting process.

**Figure 7 materials-17-01248-f007:**
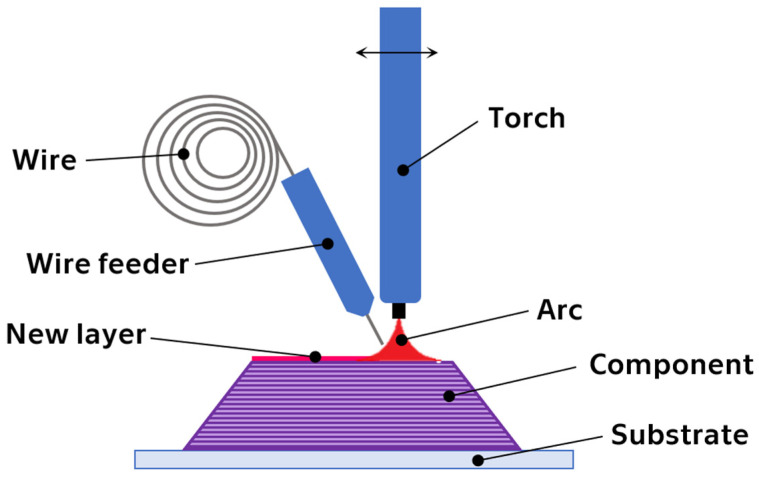
Schema of the wire arc additive manufacturing process.

**Figure 10 materials-17-01248-f010:**
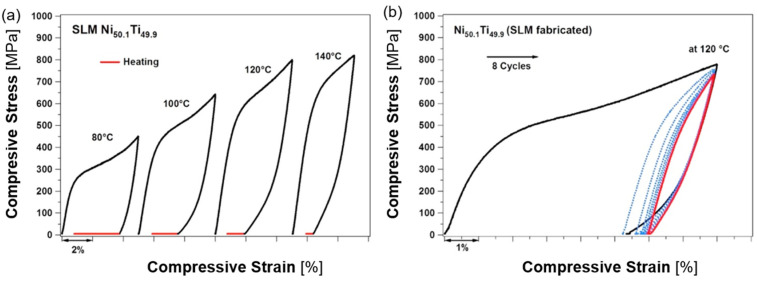
(**a**) Temperature-dependent in compression (stress–strain) of dense PBF-LB Ni50.1Ti49.9 sample; (**b**) superelasticity cycling of dense PBF-LB Ni50.1Ti49.9 sample [23].

**Figure 11 materials-17-01248-f011:**
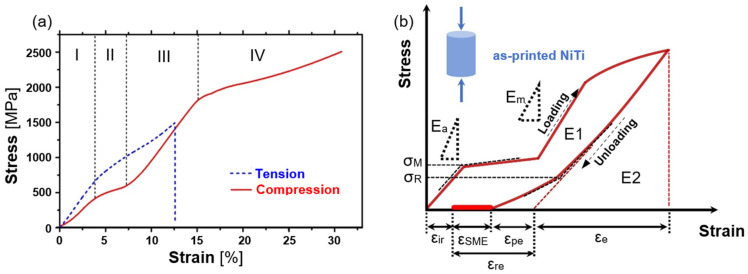
(**a**) Static compressive and tensile stress–strain curves of the as-printed NiTi (51.3 at.% Ni and 48.7 at.%) sample produced by PBF-EB, stage I—austenite elastic deformation, stage II—austenite plastic deformation or austenite to martensite transformation and martensite detwinning, stage III—detwinned martensite elastic deformation, stage IV—martensite plastic deformation; (**b**) schematic of cyclic compression (*ε_ir_*, *ε_SME_*, *ε_pe_*, ε*_re_*, and *ε_e_* denote irreversible strain, shape memory effect strain by heating above *A_f_*, pseudoelastic strain, recoverable strain, and elastic strain, respectively; *E*_1_ is dissipated heat energy and *E*_2_ is the total energy available upon unloading) of the as-printed NiTi (51.3 at.% Ni and 48.7 at.%) sample produced by PBF-EB [104].

**Figure 12 materials-17-01248-f012:**
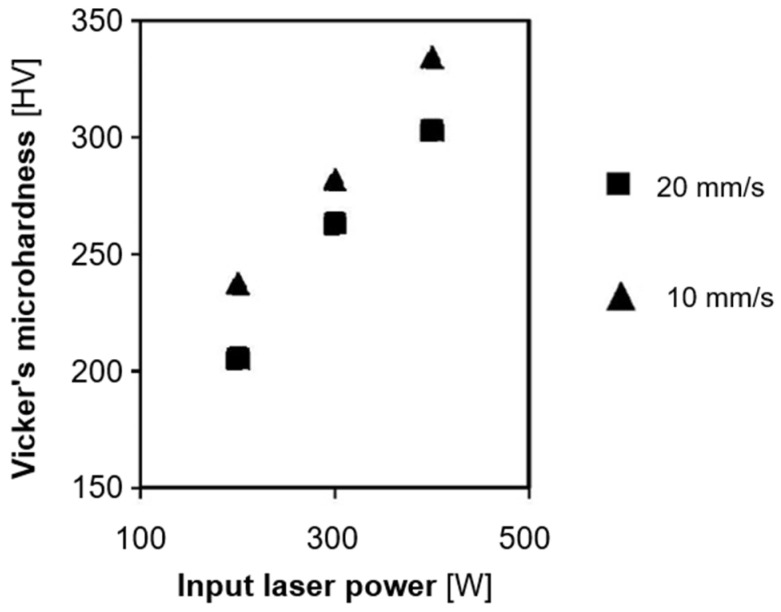
The effect of DED-LB processing parameters—Vickers’s microhardness [10].

**Figure 13 materials-17-01248-f013:**
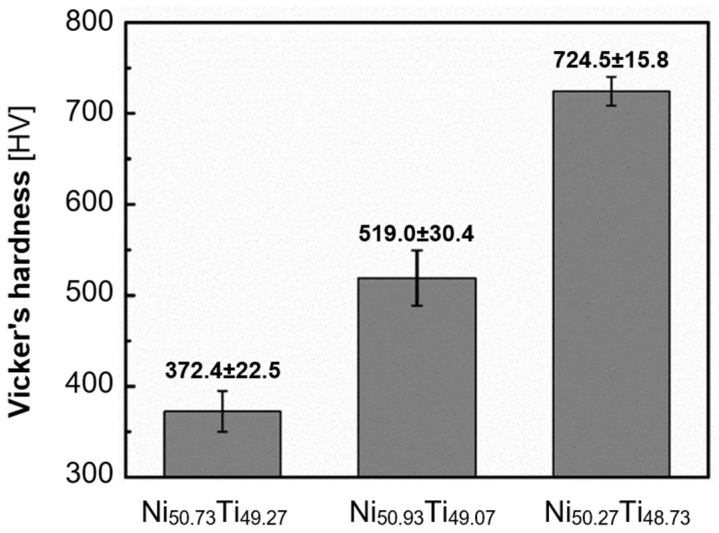
Vickers hardness of PBF-LB NiTi samples with different Ni contents [65].

**Figure 14 materials-17-01248-f014:**
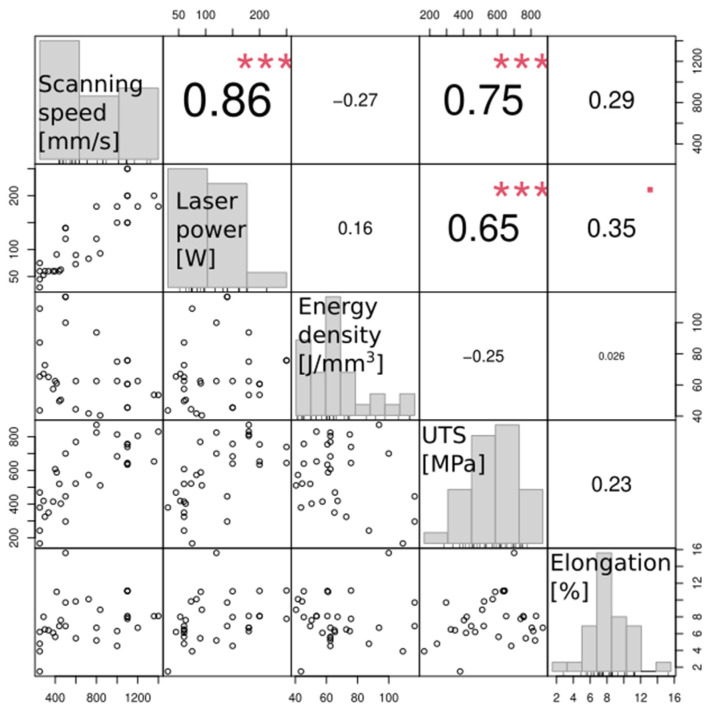
Correlation matrix between processing and mechanical properties of NiTi alloy produced by PBF-LB. The distribution of each variable is shown on the diagonal. The scatter plots are displayed at the bottom of the diagonal, while the value of the Pearson correlation plus the significance level (as stars) is shown above the diagonal line. Each significance level is associated with a symbol: *p*-values 0.001, and 0.1 correspond to symbols ***, ^▪^, respectively).

**Figure 15 materials-17-01248-f015:**
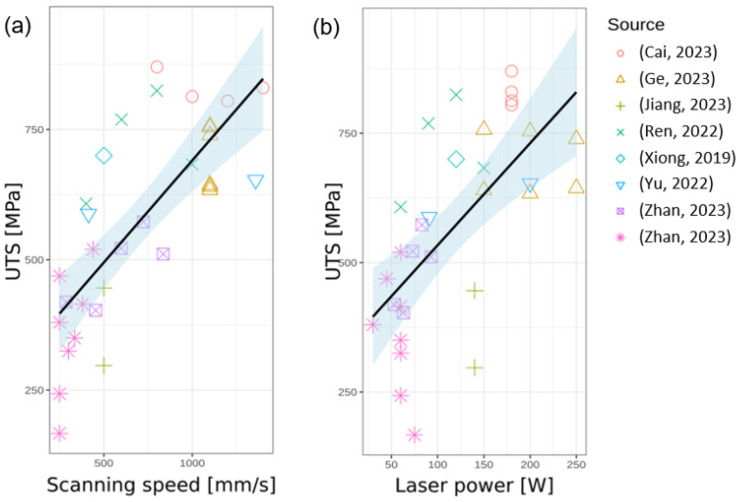
Relationship between ultimate tensile strength (UTS) and two key process parameters in PBF-LB additive manufacturing: (**a**) scanning speed and (**b**) laser power. Each point represents the UTS value from various studies indicated by point shape. Black line shows linear fit, while shaded areas depict the corresponding 95% confidence intervals [20,35,41,54,55,57,60,114].

**Figure 16 materials-17-01248-f016:**
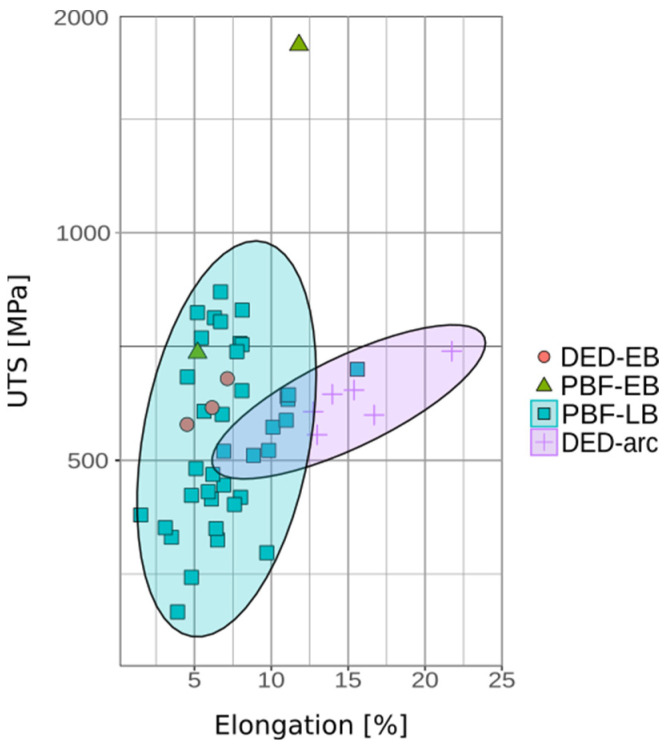
Overview of ultimate tensile strength and elongation of AM NiTi alloy obtained from various additive manufacturing techniques. Concentration ellipses are shown for applicable data sets.

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
