# Peer review of "A Review on Additive Manufacturing Methods for NiTi Shape Memory Alloy Production"

_materials, 2024, doi:10.3390/ma17061248_

Round 1

Reviewer 1 Report

Comments and Suggestions for Authors

In this manuscript, five additive manufacturing methods for fabricating NiTi alloy are reviewed, encompassing their fundamental principles, advantages/disadvantages, and the influence of input parameters on the mechanical properties, hardness, shape memory effect, and superelastic performance of NiTi alloy. This manuscript provides a comprehensive overview of additive manufacturing methods for NiTi ally and helps to promote academic progress and knowledge dissemination. However, several holes undermine the integrity of the report. The following issues would need to be addressed or clarified. 

1.       The major issue of this review article is the fourth section, AM technology dependent properties. It simply lists the literature without accurately presenting them through rational logic, making it impossible to untangle the relationships among existing research results for readers. Most importantly, the author's perspective is notably absent. Furthermore, it should closely revolve around the research question, and the language used should be more brief and clear.

2.       In the title, “memory shape NiTi alloy” should be NiTi shape memory alloy.

3.       In the Introduction of this review article, the historical development and unique properties (shape memory alloy, superelasticity and biocompatibility) of NiTi alloy are expounded in the first seven paragraphs. However, these details are excessively verbose and not central to the main focus. It is recommended to abbreviate these details or present them as a separate section.

4.       In Figures 1,10,11 and 12, the labels 'a', 'b' and 'c' are missing.

5.       On page 18, the sentence “a very good tensile strength of 42.5 GPa was achieved” contradicts the information presented in reference 52."

6.       Table 3 should have additional descriptions

7.       There should be a comparison among the five additive manufacturing processes to determine which method is more effective for fabricating NiTi alloy.

8.       This manuscript lacks a prospect on the additive manufacturing methods for NiTi alloy.

Reviewer 2 Report

Comments and Suggestions for Authors

Thanks to the authors for submitting their manuscript titled «A review on additive manufacturing methods for memory shape NiTi alloy production» to the journal «Materials».

This article confidently assesses the relevance, scientific novelty, and practical significance of additive manufacturing methods for nickelide-titanium alloys with shape memory effect. These alloys possess unique properties and are widely used in medicine and other fields. Additive technologies offer a highly promising opportunity for producing complex-shaped products from these alloys. This work systematises and generalises the latest data on additive manufacturing of nickelide-titanium alloys, including the influence of process parameters on microstructure and properties. The results have practical significance, as they can be used to optimise the processes of additive manufacturing of nickelide-titanium products with specific properties.

Comments:

1. Additive technology belongs to the cutting edge of modern science and technology. Over the past year a very impressive volume of review publications by various authors around the world on this subject has been published. It makes sense to list them. In particular, the literature review does not contain enough up-to-date sources for 2023 (only 4 sources are listed). It would be useful to supplement the literature list with recent publications in this field. In general, a review article of 24 pages for less than 100 sources looks very modest. Please make a comparative analysis between traditional sintering methods such as, for example, SPS, HIP, CIP+ hot press, CIP+ MW, vacuum sintering. Including, there is no comparison of NiTi composition with other intermetallides such as TiCu [10.3390/met12071089].

2. The section on methods for investigating the microstructure and physical and mechanical properties seems incomplete. It is worth describing in more detail each of the methods used and the principles of their operation. The paper completely ignores the issue of phase formation during the preparation of NiTi alloys.

3. It would be expedient to present a summary table of obtained mechanical characteristics of NiTi for different AM methods with indications of optimal production modes at the end of the paper. This would allow a visual comparison of the efficiency of the methods.

In conclusion, the reviewed paper addresses an important topic and will undoubtedly be of great interest to researchers in the field of additive manufacturing technologies for materials with special functional properties. The relevance of the topic, scientific novelty, and practical significance of the work are beyond question. While the paper does have some typical shortcomings of review papers, such as fragmentation, repetitions, and stylistic and grammatical errors, these do not detract from the overall quality and importance of the work. The article requires significant improvement and expansion in several sections, as well as correction of technical errors such as unifying units of measurement and adding missing elements to the article's structure. Once these issues have been addressed, the article will be suitable for publication. It is recommended that a major revision be undertaken.

Round 2

Reviewer 1 Report

Comments and Suggestions for Authors

In this revised manuscript, the authors have effectively addressed the majority of the previous issues. I can recommend accepting the paper in this journal, but not in its current form, as I still have some questions.

The detailed statistical comparison between various additive manufacturing techniques and the prospect of additive manufacturing for NiTi alloy should be treated as a separate paragraph rather than being added to the Conslusion section.

Furthermore, in line 451, Ni4Ti3 should be Ni4Ti3.

Reviewer 2 Report

Comments and Suggestions for Authors

The authors made corrections and additions to the manuscript. The paper can be published in its current form.
